# Factors Influencing Abdominal Compliance during CO_2_ Insufflation in Patients Undergoing Laparoscopic Abdominal Surgery

**DOI:** 10.3390/reports7030052

**Published:** 2024-06-28

**Authors:** Ezgi Yıldırım, K. Sanem Cakar Turhan, Aysegul Güven, Derya Gökmen, Menekse Özcelik

**Affiliations:** 1Department of Anesthesiology and ICM, Faculty of Medicine, Ankara University, Ankara 06100, Türkiye; yildirimmezgi@gmail.com (E.Y.); ayguven@ankara.edu.tr (A.G.); mozcelik@ankara.edu.tr (M.Ö.); 2Department of Anesthesiology, Yayladagı State Hospital, Hatay 31550, Türkiye; 3Department of Biostatistics, Faculty of Medicine, Ankara University, Ankara 06290, Türkiye; oztuna@ankara.edu.tr

**Keywords:** abdominal compliance, laparoscopic surgery, intra-abdominal pressure

## Abstract

The aim of this study was to investigate the effect of patient demographic and anatomical characteristics on abdominal compliance (AC), which represents the slope of the pressure–volume (P–V) curve of the abdominal cavity and is a measure of the ease of abdominal dilatation. The study included 90 patients undergoing laparoscopic abdominal surgery. Subcutaneous adipose tissue and abdominal muscle thickness were measured using ultrasonography. The mean AC was calculated during insufflation using the formula (ΔV/ΔP). The relationship between demographic and anatomical variables and AC was investigated. The results demonstrated that AC was higher in men, the elderly, and women with a history of pregnancy, and lower in patients with a history of abdominal surgery. No significant correlation was found between AC and BMI, abdominal muscle thickness, and subcutaneous adipose tissue thickness. These findings suggest that morbid obesity, a common comorbidity in laparoscopic surgery, and well-developed abdominal muscles are not indicators of low AC. However, gender, age, pregnancy history, and previous abdominal surgery affect AC during laparoscopic surgery. By taking these factors into account during preoperative evaluation, it may be possible to predict patients with low AC, which could improve perioperative outcomes through the application of individualized intra-abdominal pressure (IAP) during pneumoperitoneum.

## 1. Introduction

Carbon dioxide (CO_2_) gas is insufflated into the abdominal cavity during laparoscopic surgery to elevate the intra-abdominal pressure (IAP) for the creation of a safe and sufficient working space for the operation. The extent of insufflation varies, depending on the characteristics of the operation and the patient, and care must be taken when assigning the value, as high IAP can cause organ hypoperfusion and ischemia, leading to postoperative complications such as acute kidney injury [1]. In addition to complications, postoperative pain scores, incidence of postoperative nausea and vomiting (PONV), pulmonary complications, and length of hospital stay are also higher in laparoscopies carried out under high pressure than in those under low pressure [2,3,4,5]. In this regard, an appropriate pressure should be identified, based on a balance between ease of surgery and patient outcomes.

The elastic behaviour of the abdominal cavity determines the shape of the pressure–volume (P–V) curve of the laparoscopic workspace [6]. The slope of the P–V curve indicates the ease of abdominal dilatation and is referred to as abdominal compliance (AC), being the change in intra-abdominal volume (IAV) caused by a single unit change of pressure in the IAP [1]. The abdominal compartment is surrounded by rigid structures cranially, caudally, and posteriorly, and so AC is primarily under the influence of the elasticity of the structures comprising the abdominal wall and to a lesser extent, the diaphragm. The increase in abdominal compartment volume occurs in three phases. In the initial phase of increasing IAV, referred to as ‘*reshaping*’, IAP undergoes a minimal increase, and the shape of the abdominal cavity changes from elliptical to spherical. This change in shape is primarily due to an increase in the anteroposterior diameter and a decrease in the transverse diameter of the internal abdominal perimeter. In the second phase, designated ‘*stretching*’, the rectus abdominis muscle is elongated. Finally, in the third phase of IAV increase, designated ‘*pressurization*’, minimal increases in IAV result in a significant increase in IAP. This phase represents the exponential phase of the P–V curve [7,8]. The P–V curve, therefore, is non-linear, and the contribution of commonly used pneumoperitoneum pressures of 12–14 mmHg to IAV is limited. Consequently, AC can thus be used to predict the IAV that can be achieved with a constant target pneumoperitoneum pressure [5,6,9,10].

The literature contains studies investigating the effect of factors such as age, subcutaneous adipose tissue thickness, abdominal circumference, muscle relaxant application, history of previous surgery, comorbid diseases, and pre-tension of the abdominal wall on AC in intensive care patients, laparoscopic procedures, and animal models, adopting different AC measurement techniques or methodologies [11,12,13,14]. For the determination of AC, it is first necessary to measure the change in IAP alongside the increase or decrease in IAV. This process occurs during the insufflation phase in laparoscopy procedures. In the present study, we investigate the effects of variables such as age, sex, American Society of Anesthesiologists (ASA) score, BMI, histories of previous surgery and pregnancy, and subcutaneous adipose tissue, rectus muscle, and lateral abdominal muscle group thicknesses on AC.

## 2. Materials and Methods

The current single-centre prospective clinical study protocol was approved by the Human Research Ethics Committee of Ankara University Faculty of Medicine (Decision no: İ07-432-22). The study protocol was registered at clinicaltrials.gov with the identification number NCT06396351. The study was conducted between August 2022 and November 2022. Written informed consent was obtained from all participants during the preoperative evaluation, and the study was conducted in accordance with the Declaration of Helsinki [15]. A sample size of 84 was calculated, based on the 0.3 correlation between BMI and CAB identified in the pilot study, with a significance level of 0.05 and a power of 0.80. G*Power 3.1.9.2 was used for the sample calculation.

### 2.1. Study Population

The study included 90 patients, aged between 18 and 82 years, with ASA class I–III who underwent elective abdominal surgery using the laparoscopic technique.

### 2.2. Study Protocol

The age, sex, BMI, ASA score, number of previous abdominal surgeries, and number of pregnancies reaching the third trimester in the female patients were recorded during the preoperative evaluation of the patients. All patients fasted the night before surgery, in accordance with ASA guidelines, and were premedicated with 1 mg of midazolam and 50 mcg of fentanyl before surgery [16]. Three-lead electrocardiogram (ECG), peripheral O_2_ saturation (SPO_2_), noninvasive blood pressure (NIBP), and train-of-four (TOF) monitoring were performed before induction of general anaesthesia. Subcutaneous adipose tissue and abdominal muscle thickness were measured by ultrasound (Venue 40, GE Medical Systems, Wuxi, China), and the total thickness of the external oblique, internal oblique, and transversus abdominis muscles was measured and recorded as the lateral abdominal muscle group. All measurements were performed by the same investigator to ensure measurement reliability, with three consecutive measurements carried out for each variable, the mean value of which was calculated for use in the study. During the induction of general anaesthesia, 2–3 mg/kg propofol and 1 mg/kg rocuronium were administered, and endotracheal intubation was performed after a TOF value of 0 was reached. Mechanical ventilation (Drager Perseus A500, Lübeck, Germany) was provided in volume-controlled mode with FiO_2_ 40%, tidal volume 6 mL/kg (ideal weight), respiratory rate 12–14, and end-expiratory positive pressure (PEEP) 5 cmH_2_O. General anaesthesia was maintained with 4 L/min fresh gas flow and sevoflurane (1.5–2% volume) inhalation. Oropharyngeal or nasopharyngeal gastric decompression and bladder catheterization were performed before pneumoperitoneum.

All patients were kept in the supine position during the insufflation period, during which no interventions were performed by the surgical team. The insufflator (UHI-4, Olympus Medical Systems Corp., Tokyo, Japan) was set to an IAP of 15 mmHg and a flow rate of 4 L/min, with the value appearing on the insufflator at the beginning of the insufflation being recorded as the basal IAP.

The insufflation procedure was divided into three phases, and recorded as follows:Phase 1: when IAP reached 5 mmHg from baseline.Phase 2: when IAP had increased from 5 mmHg to 10 mmHg.Phase 3: when IAP increased from 10 mmHg to 15 mmHg.

The CO_2_ volumes insufflated to achieve an intraabdominal pressure of 5 mmHg (IAV_1_), 10 mmHg (IAV_2_), and 15 mmHg (IAV_3_) were recorded (Figure 1).

Abdominal compliance values were calculated for each phase by dividing the volume of CO₂ insufflated during that phase (ΔV) by the change in IAP that occurred during that phase (ΔP). The mean abdominal compliance of the entire insufflation process, until the patient’s intra-abdominal pressure increased from baseline to 15 mmHg, is presented as AC, while the abdominal compliance of Phase 1 is presented as AC_1_, the abdominal compliance of Phase 2 as AC_2_, and the abdominal compliance of Phase 3 as AC_3_. The calculations are based on the following formulae:
$$AC(mL/mmHg)=\frac{\Delta V}{\Delta P}=\frac{IAV3(mL)}{15-basal\;IAP(mmHg)}$$
$$AC1(mL/mmHg)=\frac{\Delta V1}{\Delta P1}=\frac{IAV1(mL)}{5-basal\;IAP(mmHg)}$$
$$AC2(mL/mmHg)=\frac{\Delta V2}{\Delta P2}=\frac{IAV2-IAV1(mL)}{10-5(mmHg)}$$
$$AC3(mL/mmHg)=\frac{\Delta V3}{\Delta P3}=\frac{IAV3-IAV2(mL)}{15-10(mmHg)}$$

### 2.3. Statistical Analysis

IBM SPSS Statistics for Windows (Version 20.0. Armonk, NY, USA: IBM Corp.) was used for the statistical analysis of the study. Sample numbers and percentages were used for categorical data, while mean, standard deviation, median, 25–75% quartiles, and minimum and maximum values were used for continuous data. A normality analysis of the data was performed visually using a Shapiro–Wilk test and histogram and QQ-plot graphs. For the mean/median comparisons of two independent groups, a Mann–Whitney U test was used for non-normally distributed parameters, and an independent samples-*t* test was employed for normally distributed parameters. For the mean/median comparisons of more than two independent groups, a Kruskal–Wallis test was used for non-normally distributed parameters, and a one-way ANOVA test was employed for normally distributed parameters. For the evaluations involving multiple groups, the *p*-value for the subgroup analyses was evaluated with a Bonferroni correction. A Wilcoxon signed-rank test was used for the comparison of two dependent groups with non-normally distributed variables, and a Spearman correlation test was employed for the correlation analysis of non-normally distributed variables. A ROC analysis was performed to determine the cut-off value for continuous data between the two groups. *p* < 0.05 was set as the statistical significance level.

## 3. Results

The study was completed with 90 patients, including 39 (43.3%) males and 51 (56.7%) females. All our patients were monitored during insufflation, and we did not observe any adverse effects that could suggest hyperinflation, such as sudden cardiac arrest, gas embolism, arrhythmia, or death. The mean age of the patients was 54 ± 14 years, and the mean BMI was 27.1 ± 4.5 kg/m^2^ (Table 1).

BMI was analysed in six groups, and the patient distribution is shown in Figure 2.

In terms of pregnancy history, the study included 51 female patients. Of these, 47 (92.2%) had a pregnancy history, while only 4 (7.8%) had never experienced a pregnancy reaching the third trimester (Figure 3). The mean number of pregnancies in the female patients was 2 ± 2.

In terms of previous abdominal surgery, 31 (34.4%) out of the 90 patients had undergone a range of previous abdominal surgeries, whereas 59 (65.6%) patients had no history of abdominal surgery (Figure 4). The mean number of previous abdominal surgeries in all patients was 1 ± 1.

The mean thicknesses of the subcutaneous adipose tissue, rectus abdominis muscle, and lateral abdominal muscle group in the sample were 2.74 ± 1.17, 1.04 ± 0.27, and 1.86 ± 0.5 cm, respectively. The total volume of CO_2_ insufflated to bring the IAP to 15 mmHg (IAV_3_) was 3979 ± 1085 mL. The mean AC was 308 ± 84 mL/mmHg, while AC_1_, AC_2_, and AC_3_ were 443 ± 196, 334 ± 118, and 209 ± 73 mL/mmHg, respectively (Table 2).

The volume measurements at each of the three pressure points for patients and the P–V curve of the abdominal cavity are detailed in Figure 5.

AC was significantly higher in the male patients (337 ± 96 mL/mmHg) than in the female patients (286 ± 65 mL/mmHg) (*p* < 0.05) (Table 3).

In an evaluation of the relationship between AC and ASA scores, a statistically significant difference was observed only for AC_1_ (*p* < 0.05). A subgroup analysis revealed the difference in both parameters to be attributed to the fact that the number of patients in ASA class I was significantly lower than the number of patients in ASA class II (Table 4).

A weak positive correlation was noted between AC, AC_1_, AC_2_, and age, but no correlation was noted between AC and BMI. When women with at least one pregnancy reaching the third trimester were compared with those with no history of pregnancy, AC and AC_2_ were noted to be significantly higher in the group with a history of pregnancy (Mann–Whitney U test, *p* < 0.05). When the correlation relationship between the number of pregnancies and AC was examined in women with a history of pregnancy, a positive correlation was found with AC, AC_1_, and AC_2_ parameters, with the AC and AC_1_ parameters being recorded as significantly lower in the group with a history of at least one abdominal surgery compared to the group with no such history (independent samples *t*-test *p* < 0.05). Similarly, a weak negative correlation was found between the number of abdominal surgeries and AC and AC_1_. No correlation was found between AC and BMI, and similarly, no correlation was found between BMI groups and AC (r_s_ = 0.111, *p* = 0.299) (Table 5).

An analysis of the relationship between AC and abdominal muscle thickness revealed a weak negative correlation between rectus abdominis muscle thickness and AC_1_, while no statistically significant correlation was found with the thickness of the lateral abdominal muscle group. Finally, a weak negative correlation was found between subcutaneous adipose tissue thickness and AC_1_ (Table 6).

## 4. Discussion

The mean AC of the patients in the present study was found to be 308 ± 84 mL/mmHg, and AC decreased from the first to the third phase, compatible with the reshaping, stretching, and pressurization phases described in the literature. The pressurization phase, corresponding to the exponential period in the intraabdominal pressure–volume curve, occurs after the IAP rises above 15 mmHg, although in patients with low AC and a high initial IAP, the pressurization phase may start at a much lower IAP during insufflation [1,7]. In the present study, IAP and AC measurements were performed at three points, and as the P–V curve could not be generated, it cannot be determined that phase 3 in the present study represented the pressurization phase for every patient.

One of the most important findings of the present study was that no correlation was found between BMI and AC, nor between the thickness of the subcutaneous adipose tissue and AC. A review of literature related to this subject revealed the resting IAP of the obese patient group to be higher, while AC is not correlated with BMI, suggesting that body composition and adipose tissue distribution may be important factors [7,17,18]. People with an “apple or android body” morphology, known also as central obesity, have more visceral adiposity, a spherical abdominal shape, and higher baseline BMI and IAP than those with a “pear or gynoid body” morphology with peripheral adiposity, and so are thought to have limited reserves for additional IAV and lower AC [1]. In this regard, it may be more appropriate to assess body morphology along with BMI for the estimation of AC.

Another important finding of the study was the lack of any correlation identified between abdominal muscle thickness and AC. In theory, the muscle layer of the abdominal wall should play an important role in determining the slope of the P–V curve; however, no such effect was identified in the present study [7]. Muscle thicknesses were measured while the patients were awake, while AC measurements were performed under general anaesthesia followed by deep muscle blockade, and it can thus be suggested that the potential attenuating effect of abdominal muscles on AC will vary according to the level of neuromuscular blockade and may even disappear under complete blockade. Dillemans et al. measured the thickness of the rectus muscle ultrasonographically at the level of the umbilicus after complete muscle relaxation under general anaesthesia in their study of 20 obese patients undergoing laparoscopic gastric bypass surgery. Similar to the present study, they evaluated the relationship with the IAV required to bring the IAP to 15 mmHg, but identified no correlation between insufflated volume and rectus muscle thickness [19].

There are, however, conflicting reports in the literature regarding the relationship between the level of neuromuscular blockade and AC. Chassard et al., for example, identified no difference in AC between the groups with and without neuromuscular blockade during insufflation in their study of pigs [11]. Conversely, Danneels et al. found that muscle relaxation increased IAV in their study of 33 ASA class I or II patients undergoing bariatric laparoscopic intervention [20]. Finally, Barrio et al. investigated the effect of deep (TOF = 0) and moderate (TOF = 1–3) neuromuscular blockade on AC in patients undergoing laparoscopic surgery with different target intraabdominal pressure levels and reported AC to be higher in those with deep neuromuscular blockade when compared to those with moderate neuromuscular blockade at both the 8 mmHg and 12 mmHg IAP values. In the present study, AC measurements were obtained at both the 8 mmHg and 12 mmHg IAP levels in patients with moderate neuromuscular blockade, after which desufflation was performed, followed by deep neuromuscular blockade induction in all patients. While under deep neuromuscular blockade, the patients underwent insufflation a second time, and the AC measurements were repeated; thus, the increased AC may be attributed to the effect of the initial pneumoperitoneum, which may cause a relaxed abdominal wall, like a ‘recruited lung’ [21].

The results of the present study reveal that AC increases with increasing age, although the weak correlation observed may limit the clinical use of this finding, as previous studies in the literature have reported a decrease in AC with advancing age [22,23]. The loss of viscoelastic properties in the abdominal wall and diaphragm as age progresses could potentially contribute to an increase in AC, particularly during the reshaping and stretching phases, and the greater incidence of previous surgeries among elderly individuals may further influence this finding.

Mulier et al. reported no difference between AC and gender in their study of 70 patients with ASA class I or II who underwent laparoscopic surgery. As there are very few articles examining the relationship between AC and gender in the literature, it is worth noting that AC was found to be higher in the male patients than in female patients in the present study [24].

An analysis of the relationship between ASA score and AC revealed the AC_1_ parameter to be statistically significantly lower in the ASA I patients than in the ASA II patients, while there were no significant findings related to other AC parameters or ASA III patients. In previous studies in the literature, comorbidities such as hepatomegaly, splenomegaly, ascites, COPD, and extensive abdominal burns are associated with decreased AC [7]. There has been no study to date evaluating the relationship between ASA score and AC. A difference may not have been detected due to the lack of patients with ASA IV and V in the present study, the unequal distribution of patients between the ASA I-II-III groups, and the fact that only 16% of patients were categorized as ASA III.

A positive correlation was noted in the present study between the number of pregnancies in the female patients and AC, and a comparison of the female patients with and without a history of at least one pregnancy revealed a significantly higher AC in the group with a history of pregnancy. Conversely, a negative correlation was noted between the number of previous abdominal surgeries and AC. In a comparison of the patients with a history of at least one abdominal surgery with those with no such history, AC was found to be significantly lower in the group with a history of abdominal surgery. In a study by Verbeke et al., an increase in AC was noted in the female patients during laparoscopic gynaecologic surgery, although the increase was lower in those with a history of multiple pregnancies, laparoscopy, and laparotomy for each group, separately, and the AC measurements at the beginning of insufflation were higher in the same group of patients [25].

In another prospective study evaluating the relationship between parity and the volume of CO_2_ insufflated to achieve constant pressure targets in female patients undergoing laparoscopic gynaecologic surgery, Abu-Rafea et al. found parity to be positively correlated with IAV for all pressure slices (also correlated with AC) [26]. A comparison of the results of our study with those of the above studies suggests that a history of pregnancy, as a factor in increased chronic intra-abdominal pressure, may increase AC by causing permanent damage to the viscoelastic components of the abdominal wall. Similar to pregnancy, laparoscopic surgery may also increase AC by causing loss of viscoelasticity in the abdominal wall. As such, the negative correlation observed between previous abdominal surgery and AC in our study may be related to the fact that patients with a history of laparotomy and related intra-abdominal adhesions constituted a majority in our patient population. It can thus be considered important to differentiate between laparotomy and laparoscopy when evaluating the association between previous abdominal surgeries and AC.

The present single-centre study has several limitations. Firstly, our approach to the evaluation of AC based on three-point measurements can be considered a limitation, as AC is a dynamic parameter representing the slope of the abdominal cavity P–V curve. Secondly, grouping laparoscopy and laparotomy as “previous abdominal surgeries” complicates the interpretation of the study results, as these procedures may have different physiopathological effects on the elastic components of the abdominal wall structures. Finally, the lack of any investigation into the causal relationship between the patients’ hemodynamic parameters and abdominal compliance can be considered a further limitation.

## 5. Conclusions

The results of the present study suggest that AC, which is a measure of the ease of expansion of the abdominal cavity and is represented by the slope of the P–V curve, is higher in men, older adults, and women with a history of pregnancy. This leads us to the conclusion that in these patient groups, adequate surgical working space can be achieved by utilizing lower insufflator pressure during pneumoperitoneum. The results of the present study concur with those in the literature in suggesting that morbid obesity, a significant concern in laparoscopic surgery, is not solely indicative of low AC. In addition, the expected reduction in AC due to muscle mass in patients with well-developed abdominal muscles is negated under deep neuromuscular blockade. The determination of the factors that influence abdominal compliance may facilitate the identification of patients with reduced abdominal compliance, thereby reducing perioperative complications through individualized IAP management during laparoscopic surgery. Our findings may also help identify patients at risk for intra-abdominal hypertension or abdominal compartment syndrome in the critical care setting of the intensive care unit.

## Figures and Tables

**Figure 1 reports-07-00052-f001:**
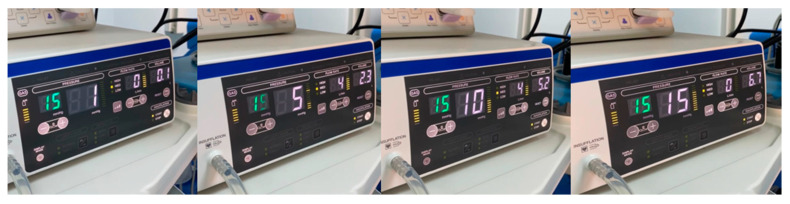
Phases of insufflation.

**Figure 2 reports-07-00052-f002:**
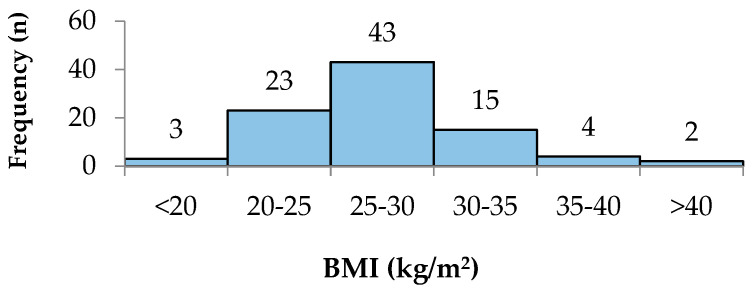
The BMI distribution.

**Figure 3 reports-07-00052-f003:**
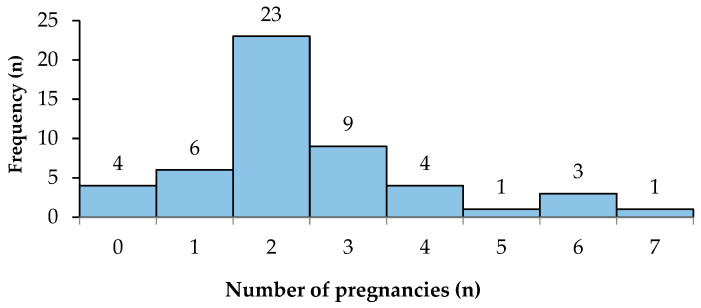
The frequency of pregnancies that reach the third trimester in female patients.

**Figure 4 reports-07-00052-f004:**
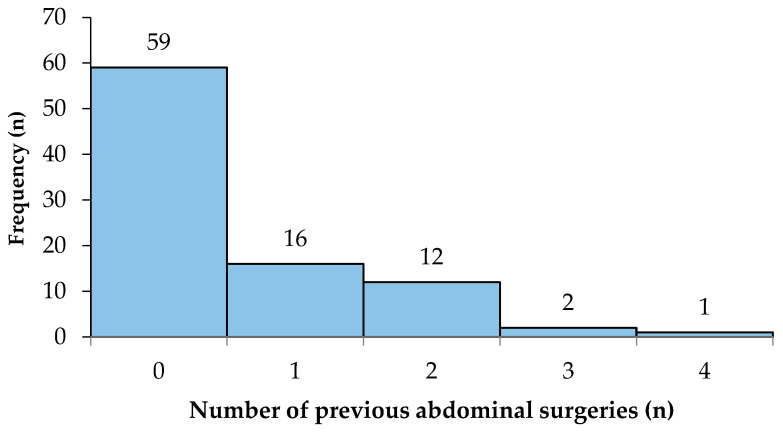
The distribution of the number of previous abdominal surgeries.

**Figure 5 reports-07-00052-f005:**
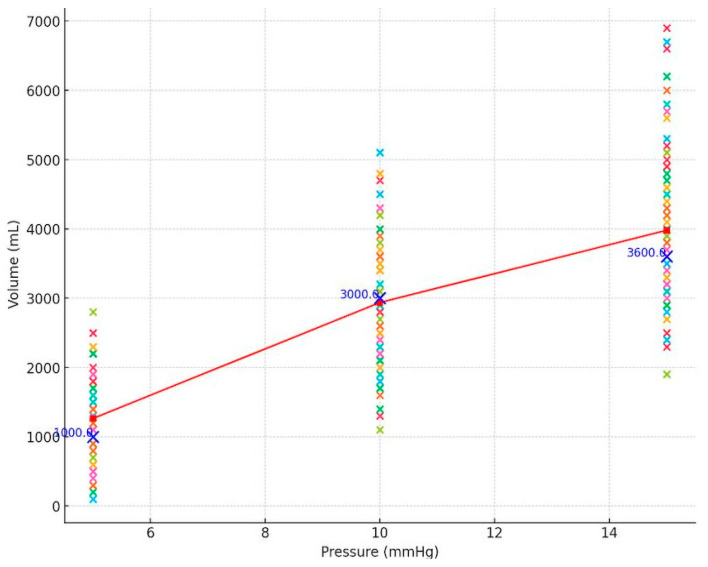
Pressure–Volume relationship of abdominal cavity during pneumoperitoneum. The graph illustrates the volume measurements at three pressure points (5, 10, and 15 mmHg) for all participants in the study. The coloured crosses represent the individual measurements. Mean volume values at each pressure point are denoted by red dots, forming a representative P–V curve. The blue crosses indicate the mode volume measurements at the corresponding pressure level.

**Table 1 reports-07-00052-t001:** Demographic data.

Gender, n (%)	
Female	51 (56.7)
Male	39 (43.3)
Age, mean ± SD	54 ± 14
Female	55 ± 13
Male	51 ± 15
ASA score, n (%)	
I	21 (23.3)
II	53 (58.9)
III	16 (17.8)
Weight, kg, mean ± SD	73 ± 12
Height, cm, mean ± SD	164 ± 9
BMI, kg/m^2^, mean ± SD	27 ± 4.5

**Table 2 reports-07-00052-t002:** Descriptive variables of the study.

	mean ± SD (median)
Number of pregnancies, n	2 ± 2 (2)
Number of previous abdominal surgeries, n	1 ± 1 (0)
Thickness of subcutaneous adipose tissue, cm	2.74 ± 1.17 (2.72)
Thickness of rectus abdominis muscle, cm	1.04 ± 0.27 (1.03)
Thickness of the lateral abdominal muscle group, cm	1.86 ± 0.5 (1.84)
Basal IAP (mmHg)	2 ± 1 (2)
IAV_3_, mL	3979 ± 1085 (3800)
AC_1_, mL/mmHg	443 ± 196 (413)
AC_2_, mL/mmHg	334 ± 118 (340)
AC_3_, mL/mmHg	209 ± 73 (200)
AC, mL/mmHg	308 ± 84 (300)

**Table 3 reports-07-00052-t003:** C_ab_—gender relationship.

	Female		Male		
	mean ± SD	median	mean ± SD	median	*p*
AC_1_, mL/mmHg	387 ± 159	366	516 ± 218	466	0.003 *
AC_2_, mL/mmHg	316 ± 101	320	358 ± 134	360	0.094 **
AC_3_, mL/mmHg	202 ± 72	200	217 ± 74	200	0.304 *
AC, mL/mmHg	286 ± 65	291	337 ± 96	323	0.007 **

* Mann–Whitney U test; ** independent samples-*t* test.

**Table 4 reports-07-00052-t004:** AC–ASA score relationship.

	ASA I		ASA II		ASA III		
	mean ± SD	median	mean ± SD	median	mean ± SD	median	*p*
AC_1_, mL/mmHg	331 ± 134	300	478 ± 208	433	473 ± 177	413	0.011
AC_2_, mL/mmHg	312 ± 84	320	348 ± 121	360	319 ± 144	300	0.515
AC_3_, mL/mmHg	211 ± 63	200	206 ± 72	200	212 ± 89	180	0.826
AC, mL/mmHg	277 ± 53	278	319 ± 85	300	313 ± 105	325	0.226

Notes: The Kruskal–Wallis test and Bonferroni correction were used for the subgroups.

**Table 5 reports-07-00052-t005:** Relationship of AC with other demographic data.

	AC		AC_1_		AC_2_		AC_3_	
	r_s_	*p*	r_s_	*p*	r_s_	*p*	r_s_	*p*
Age	0.283 **	0.007	0.366 **	<0.001	0.231 *	0.028	−0.056	0.602
BMI	0.103	0.333	−0.061	0.566	0.183	0.084	0.191	0.072
Number of pregnancies	0.472 **	<0.001	0.350 *	0.012	0.44 **	0.001	−0.02	0.887
Previous abdominal surgeries	−0.218 *	0.039	−0.266 *	0.011	−0.078	0.464	−0.119	0.265

** Correlation is significant at the 0.01 level (2-tailed); * correlation is significant at the 0.05 level (two-tailed); r_s_: Spearman correlation coefficient.

**Table 6 reports-07-00052-t006:** Relationship of AC with thickness of abdominal wall structures.

	AC		AC_1_		AC_2_		AC_3_	
	r_s_	*p*	r_s_	*p*	r_s_	*p*	r_s_	*p*
Thickness of the rectus abdominis muscle	−0.034	0.749	−0.219 *	0.038	0.091	0.393	0.103	0.333
Thickness of the lateral abdominal muscle group	0.062	0.562	−0.012	0.911	0.074	0.487	0.136	0.201
Thickness of the subcutaneous adipose tissue	−0.091	0.394	−0.263 *	0.012	0.034	0.753	0.080	0.453

* Correlation is significant at the 0.05 level (two-tailed); r_s_: Spearman correlation coefficient.

## Data Availability

The datasets generated and/or analysed during the current study are available from the corresponding author on reasonable request.

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
