# Peer review of "Factors Influencing Abdominal Compliance during CO2 Insufflation in Patients Undergoing Laparoscopic Abdominal Surgery"

_reports, 2024, doi:10.3390/reports7030052_

Round 1

Reviewer 1 Report

Comments and Suggestions for Authors

The authors assessed the abdominal wall compliance while establishing pneumoperitoneum, a basic step in laparoscopic surgery. The keys finding, amongst others are that the abdominal compliance is high in male, older patients and those women who have been pregnant before

A few queries and comments

1. The key finding relevant to clinicians is that BMI does not affect AC. As there are only 90 patients, could the absence of difference due to type 2 error, i.e. relatively small sample size?

2. Introduction paragraph 2: "shape of the abdomen changes from spherical to circular". This sentence may be difficult to comprehend to the readers. "circular' usually refers to a 2 dimensional object whereas spherical refers to 3 dimensional object. One would imagine the abdominal cavity becomes spherical with insufflation of carbon dioxide. May be the authors can explain in another way.

3. Same paragraph as above: "the IAP undergoes a significant increase, but without significant increase in IAV." Wouldn't one expect the other way around, i.e. a volume of 3979ml carbon dioxide insufflated with change of 15mmHg pressure? 

Comments on the Quality of English Language

4. the manuscript would benefit from editing by a native English speaker.

Author Response

                                                                                                                    June 17th, 2024    

Dear editorial team,

We are very grateful to you for taking the time to review our paper. We appreciate your comments and contributions, which have helped us to improve our work. Please find below our detailed responses to your feedback, as well as the relevant revisions and corrections in the resubmitted manuscript. We have also made revisions to the introduction, results and conclusion sections of the study based on the feedback of both reviewers. We have made further improvements to the visual appearance of the manuscript by adding histogram plots. We have also corrected a minor error in our terminology and removed a reference containing a different definition of 'compliance' from the manuscript after the feedback. We would like to thank our reviewers once again for their guidance.

Best Regards...

* Corresponding author: Keziban Sanem Çakar Turhan

Address: Ankara University Faculty of Medicine, Ibni Sina Hospital Department of Anesthesiology and ICU, Ankara Turkey

Tel: 00 90 533 3689520   Fax: 00 90 312 311 5057

E-mail: sanemcakar@yahoo.com       

Response to Reviewer Comments

1.      Summary

We would like to express our gratitude for taking the time to review our article. Your comments and contributions are greatly valued and have helped us improve our work. You can find below our detailed responses to your feedback, as well as the relevant revisions and corrections contained in the resubmitted file.

2.      Questions for General Evaluation

Reviewer’s Evaluation

Response and Revisions

Does the introduction provide sufficient background and include all relevant references?

Yes

No additional comments.

Are all the cited references relevant to the research?

Yes

No additional comments.

Is the research design appropriate?

Yes

No additional comments.

Are the methods adequately described?

Yes

No additional comments.

Are the results clearly presented?

Yes

No additional comments.

Are the conclusions supported by the results?

Yes

No additional comments.

3.      Point-by-point response to Comments and Suggestions for Authors

Comments 1: The key finding relevant to clinicians is that BMI does not affect AC. As there are only 90 patients, could the absence of difference be due to type 2 error, i.e. relatively small sample size?

Response 1:  We appreciate your concern regarding the potential for a type 2 error due to the sample size. We conducted a comprehensive pilot study as part of our study process. Our pilot study revealed a correlation of 0.3 between BMI and AC. Based on these findings, we determined that the sample size should be 84 with a significance level of 0.05 and 80% power. Consequently, we believe that our sample size is adequate and that we have taken the necessary measures to minimise the possibility of a type 2 error. However, as can be seen from the BMI distribution histogram on page 4, which we added with the revision, we have 4 patients with a BMI between 35-40 and only 2 patients with a BMI greater than 40. It would appear that a significant portion of the patients are clustered between BMI 25-30. Consequently, rather than the sample size, this situation is a cause for concern in terms of type 2 error. In order to reduce this possibility, we conducted a correlation analysis of BMI with AC, both with and without grouping, as we mentioned in the article.

Comments 2:  Introduction paragraph 2: "shape of the abdomen changes from spherical to circular". This sentence may be difficult to comprehend to the readers. "circular' usually refers to a 2 dimensional object whereas spherical refers to 3 dimensional object. One would imagine the abdominal cavity becomes spherical with insufflation of carbon dioxide. Maybe the authors can explain in another way.

Response 2: We would like to express our gratitude for your valuable critique, which highlighted the terminological error and ambiguity present in our manuscript.  In response to your feedback, we have revised the sentence in the second paragraph of the introduction section on page 2 as follows: 'In the initial phase of increasing IAV, referred to as 'reshaping', IAP undergoes a minimal increase and the shape of the abdominal cavity changes from elliptical to spherical. This change in shape is primarily due to an increase in the anteroposterior diameter and a decrease in the transverse diameter of the internal abdominal perimeter.’

 Comments 3: Same paragraph as above: "The IAP undergoes a significant increase but without a significant increase in IAV." Wouldn't one expect the other way around, i.e. a volume of 3979ml carbon dioxide insufflated with a change of 15mmHg pressure? 

Response 3: Thank you for your insightful feedback on the relationship between IAP and IAV. Further review of the existing literature on this topic shows that the pressurization phase is clearly defined as the exponential phase in the pressure-volume (P-V) curve of the abdominal cavity. However, the existing literature does not provide a clear cut-off value for the transition from the stretching phase to the pressurization phase, giving estimated values in a wide range of 5600 to 7600 ml. It remains unclear whether a precise cut-off value can be defined, and further studies on this subject would be beneficial. In our study of patients undergoing elective abdominal surgery, we did not cross the threshold for intra-abdominal hypertension, prioritizing patient safety by ensuring that IAP did not exceed 15 mmHg. As a result, the total insufflated CO2 volume (IAV3) in our study was 3979 ± 1085 ml. When we evaluate these results in the light of the literature, it is important to acknowledge that some patients in our study may not have entered the pressurization phase during the measurements. This represents a potential limitation of our study and highlights the need for further research to clarify the optimal IAV limit for pressurization. Nevertheless, although we did not set a definite IAV limit for pressurization, we believe that our study is valuable in terms of identifying factors affecting abdominal compliance.

Furthermore, to avoid confusion among readers regarding the pressurization phase, we have revised the sentence describing the third phase in the second paragraph of the Introduction as follows: 'Finally, in the third phase of IAV increase, designated 'pressurization', minimal increases in IAV result in a significant increase in IAP. This phase represents the exponential phase of the P-V curve’

4.      Response to Comments on the Quality of English Language

Comment: The manuscript would benefit from editing by a native English speaker.

Response: Thank you for your feedback. Once the final version of the manuscript was completed, we received an editing service and certificate in English. We also checked it again after the revisions. However, if there is a specific section you would like us to review again, we would be happy to do so.

Reviewer 2 Report

Comments and Suggestions for Authors

The article is well written and show correlation between different patient’s characteristics and their abdominal compliance understood as a relation of intraabdominal volume to pressure. The results can be useful in medical practice, in my opinion in even wider extend then authors describe, because the information about mechanical behavior of abdominal wall is important in several medical topics. I have some comment to the authors that I suggest to address in the manuscript:

1) How many patients was pregnant and how many had previous surgery?

2) Term compliance has another definition in engineering, mechanics. In the literature concerning abdominal wall both definitions are used. Some of the cited articles, also use another definition (see, eg. 11), so please be aware that these two measures are not exactly the same.  Therefore, I suggest to emphasize more that this article is about relation of volume to pressure, for example write about this in the conclusions.

3) Please write in the abstract what P, V means (this is explained in the introduction, but as I wrote in point 2, it is important that the reader knows how compliance is defined in this article.

4) The study on 90 patients is valuable. Authors perform statistical analysis, which allow them to draw interesting conclusions. Nevertheless, I suggest to extend the presented results, so that more information can be extracted by the readers:

  a) Could you show graph/s presenting P-V relation (results for 3 points on one graph) for all the patients/ gropus of patients to be able to observe the shape of the P-V curve?

  b) Could you show histograms of the outcomes to see the distribution of the outcomes?

Author Response

                                                                                                                            June 17th, 2024    

Dear editorial team,

We are very grateful to you for taking the time to review our paper. We appreciate your comments and contributions, which have helped us to improve our work. Please find below our detailed responses to your feedback, as well as the relevant revisions and corrections in the resubmitted manuscript. We have also made revisions to the introduction, results and conclusion sections of the study based on the feedback of both reviewers. We have made further improvements to the visual appearance of the manuscript by adding histogram plots. We have also corrected a minor error in our terminology and removed a reference containing a different definition of 'compliance' from the manuscript after the feedback. We would like to thank our reviewers once again for their guidance.

Best Regards...

* Corresponding author: Keziban Sanem Çakar Turhan

Adress: Ankara University Faculty of Medicine, Ibni Sina Hospital Department of Anesthesiology and ICU, Ankara Turkey

Tel: 00 90 533 3689520   Fax: 00 90 312 311 5057

E-mail: sanemcakar@yahoo.com       

Round 2

Reviewer 1 Report

Comments and Suggestions for Authors

The authors have made significant amendment to improve the readability of the manuscript. The sticky points have been clarified. 

Author Response

Dear reviewer

I would like to express my gratitude for your feedback. I have incorporated the necessary revisions based on your comments and have completed the manuscript. I understand that you do not require any further revisions. Thank you for your attention and support.

Best regards,